# Respiratory Physiology of European Plaice (*Pleuronectes platessa*) Exposed to *Prymnesium parvum*

**Heiðrikur Bergsson [1],\*** , **Nikolaj Reducha Andersen [2]**, **Morten Bo Søndergaard Svendsen [1]**, **Per Juel Hansen [1]** and **John Fleng Steffensen [1]**

[1] Marine Biological Section, Department of Biology, University of Copenhagen, Strandpromenaden 5, 3000 Elsinore, Denmark; mbssvendsen@bio.ku.dk (M.B.S.S.); pjhansen@bio.ku.dk (P.J.H.); jfsteffensen@bio.ku.dk (J.F.S.)

[2] Department of Bioscience, Aarhus University, Frederiksborgvej 399, 4000 Roskilde, Denmark; nra@bios.au.dk

\* Correspondence: heidrikur.bergsson@bio.ku.dk

**Abstract:** During the last century, the blooms of the toxic haptophyte *Prymnesium parvum* have been responsible for massive fish kills in both aquaculture and wild populations. Despite decades of research, the ichthyotoxic properties of *P. parvum,* and how this alga affects fish, is still debated. Using a novel device to measure the respirometry, ventilation volume, ventilation frequency, oxygen extraction, and oxygen consumption of undisturbed European plaice (*Pleuronectes platessa)* were acquired during exposure to two algal species as well as hypoxia. Fourteen fish (258 ± 44 g) were initially exposed to severe hypoxia and left to recover for at least 48 h. Half of these fish were then exposed to known harmful concentrations of *P. parvum* (median ± standard deviation (SD); $2.6 \times 10^5 \pm 0.6 \times 10^5$ cells mL$^{-1}$), while the remaining half were exposed to the non-toxic alga *Rhodomonas salina* (median ± SD; $3.2 \times 10^5 \pm 0.7 \times 10^5$ cells mL$^{-1}$). During exposure to severe hypoxia, all of the fish were able to maintain oxygen consumption by increasing the ventilation volume. The results from fish that were exposed to *P. parvum* showed a significant decrease in oxygen extraction (median ± SD; 52.6 ± 6.9 percentage points) from pre-exposure to the end of the experiment, as opposed to fish exposed to *R. salina*, which were unaffected. These results indicate that suffocation affects the European plaice when exposed to *P. parvum*. The observed severe decrease in oxygen extraction can be ascribed to either damage of the gill epithelia or increased mucus secretion on the gills, as both would limit the transfer of oxygen, and both have been observed.

**Keywords:** *Pleuronectes platessa*; *Prymnesium parvum*; oxygen extraction efficiency; harmful algal blooms; ichthyotoxic algae; ventilatory requirement; ventilation

## 1. Introduction

The toxic alga *Prymnesium parvum* Carter has during the last century caused massive fish kills in both aquaculture and wild populations [1–5]. When a bloom of *P. parvum* occurs, all of the fish species located in the bloom area are affected [5,6]. Therefore, blooms of *P. parvum* will have ecological effects on fish stocks, as well as being a threat to aquaculture [7]. The current understanding of the ichthyotoxic mode of action of *P. parvum* is partly reversible osmotic lysis of the gill membranes [8,9]. It has been hypothesized that exposure to *P. parvum* causes the suffocation of the fishes [9]. Macroscopic investigation of the gill cell epithelia of the minnow *Gambusia affinis* shows damages during exposure to toxin(s) that were extracted from *P. parvum* [10]. The causative agents for gill damage are debated [7,11–19], and it is unknown whether a single toxin is responsible for the

membrane effects of *P. parvum* [19,20]. Damage of the gill epithelial cells will lead to reduced oxygen extraction [21]. Further, the production of mucus on the gills [7,22–24] can also cause a decreased oxygen extraction [22] due to an increased boundary layer and longer diffusion distance. This study aimed to test how exposure to ecological relevant concentrations of *P. parvum* affects physiological respiratory responses, such as oxygen extraction ($E_{O_2}$), ventilatory flow ($V_f$), ventilation frequency ($V_{freq}$), and oxygen consumption ($\dot{M}O_2$) of European plaice (*Pleuronectes platessa*) as compared to fish in normoxia and fish that are exposed to severe hypoxia under the assumption that the gills are the primary target for aquatic toxins in fishes [9,25–27]. The respiratory parameters were measured using a novel device in order to acquire measurements from non-stressed fish, which utilizes the natural behaviour of flatfish.

## 2. Results

### 2.1. Fish Groups

The body masses of the fish in the three groups were normally distributed (Shapiro–Wilk test): *P. parvum* (W = 0.91, $p$ = 0.41), *R. salina* (W = 0.91, $p$ = 0.38), and hypoxia (W = 0.95, $p$ = 0.68). The data for all of the groups did also show homoscedasticity; *P. parvum* (F = 3.5, $p$ = 0.15), *R. salina* (F = 2.36, $p$ = 0.32), and hypoxia (F = 1.48, $p$ = 0.62). With both criteria satisfying the one-way ANOVA assumptions, this test was used, and the results showed no significant difference between the masses of the fish groups (F(2,19)) = 0.52, $p$ = 0.60).

### 2.2. Exposure to Severe Hypoxia

The hypoxia group represented fish from both the *P. parvum* and *R. salina* exposure groups. Data for the respiratory parameters in normoxia and hypoxia were not normally distributed (Shapiro–Wilk test), with W-values ranging from 0.82 to 0.98 and all $p$ < 0.001. Since the data was not normally distributed, a Wilcoxon signed rank test was used to check for differences between normoxia and severe hypoxia. These differences showed that oxygen extraction significantly decreased by 15.6 ± 9.1 percentage points (median ± standard deviation (SD) (W = 52438, $p$ < 0.0001), while ventilatory flow significantly increased by 12.8 ± 3.9 mL s$^{-1}$ (median ± SD), (W = 1117, $p$ < 0.0001); this is a factor 6.1 increase. There was no significant difference between oxygen consumption (W = 29075, $p$ = 0.92), but the ventilation frequency significantly decreased by 6.3 ± 5.0 breaths min$^{-1}$ (median ± SD) (W = 45868, $p$ < 0.0001) from normoxia to severe hypoxia (Table 1, Figure 1).

**Table 1.** Summary of respiratory parameters during the different stages of each exposure. Data from post exposure to *P.* parvum is divided into two groups, the entire experiment (post exp.) and the last 10 min. before the fish left the device. A comparison between severe hypoxia and the last 10 min. of exposure to *P. parvum* are added to the bottom of the table. All of the values are given as median ± standard deviation (SD) and significance levels are shown as asterisks (*) or non-significant (ns) ($p$ > 0.05 = ns, $p$ ≤ 0.05 = *, $p$ ≤ 0.01 = **, $p$ ≤ 0.001 = *** and $p$ ≤ 0.0001 = ****).

| Exposure Treatments | Stage | $E_{O2}$ % | $V_f$ (mL × s$^{-1}$) | $V_{freq}$ (Breaths × min$^{-1}$) | $\dot{M}O_2$ (mg O$_2$ × kg$^{-1}$ × h$^{-1}$) | $V_r$ (mL × mg O$_2^-$ × kg$^{-1}$) |
|---|---|---|---|---|---|---|
| Hypoxia (n = 8) | Normoxia | 73.0 ± 6.3 | 2.8 ± 1.7 | 28.0 ± 4.8 | 20.3 ± 9.9 | 144 ± 30.4 |
| | Hypoxia | 56.2 ± 9.8 | 17.2 ± 7.7 | 19.0 ± 6.7 | 19.1 ± 9.4 | 824 ± 195 |
| | Difference | 15.6 ± 9.1 | 12.8 ± 3.9 | 6.3 ± 5.0 | 1.8 ± 8.1 | 671 ± 156 |
| | Significance | **** | **** | **** | ns | **** |
| *R. salina* (n = 7) | Pre-exp. | 74.6 ± 7.0 | 5.8 ± 3.5 | 26.0 ± 4.4 | 37.3 ± 23.8 | 146 ± 15.4 |
| | Post exp. | 74.3 ± 7.1 | 7.1 ± 4.4 | 25.0 ± 7.4 | 44.9 ± 32.3 | 140 ± 13.1 |
| | Difference | 0.3 ± 7.2 | 0.1 ± 2.3 | 1.0 ± 5.4 | 1.7 ± 13.8 | 2.2 ± 15.4 |
| | Significance | ns | ns | ns | ns | ns |

**Table 1.** *Cont.*

| Exposure Treatments | Stage | $E_{O2}$ % | $V_f$ (mL $\times$ s$^{-1}$) | $V_{freq}$ (Breaths $\times$ min$^{-1}$) | $\dot{M}O_2$ (mg $O_2 \times$ kg$^{-1} \times$ h$^{-1}$) | $V_r$ (mL $\times$ mg $O_2^{-} \times$ kg$^{-1}$) |
|---|---|---|---|---|---|---|
| *P. parvum* (n = 7) | Pre-exp. | 72.7 ± 6.9 | 2.8 ± 1.9 | 25.0 ± 5.4 | 22.0 ± 11.5 | 133 ± 30.1 |
| | Post exp. | 38.2 ± 21.1 | 5.6 ± 7.7 | 22 ± 7.9 | 23.6 ± 22.4 | 379 ± 320 |
| | Difference | 30.7 ± 16.2 | 3.4 ± 2.7 | 1.0 ± 3.2 | 4.1 ± 11.1 | 105 ± 112 |
| | Significance | **** | **** | **** | ns | **** |
| | Last 10 min | 24.0 ± 11.9 | 7.0 ± 11.5 | 19.0 ± 7.4 | 19.3 ± 29.0 | 0.4 ± 0.3 |
| | Difference | 51.7 ± 9.7 | 6.3 ± 9.3 | 4.0 ± 5.0 | 1.3 ± 25.0 | 0.3 ± 0.2 |
| | Significance | **** | **** | **** | ns | **** |
| Hypoxia vs. Last 10 min | Difference | 32.2 | 2.8 | 0 | 0.2 | 0.4 |
| | Significance | **** | **** | ns | ns | **** |

## 2.3. Algal Exposure

The exposure concentrations of *P. parvum* and *R. salina* were normally distributed (Shapiro–Wilk test): *P. parvum* (W = 0.93, *p* = 0.57), *R. salina* (W = 0.86, *p* = 0.15). The cell concentration for the two groups exhibited homoscedasticity; *P. parvum* (F = 0.8, *p* = 0.75), *R. salina* (F = 1.3, *p* = 0.75). Satisfying the assumptions of a two-sample homoscedastic t-test, a significant difference was found between the concentrations of *P. parvum* and *R. salina*; the fish were exposed to (T = 0.52, *p* = 0.046). Fish that were exposed to *R. salina* experienced a slightly higher cell concentration then fish exposed to *P. parvum*.

During exposure to *P. parvum* and *R. salina*, the fish experienced an elevated ambient pH level, due to higher pH in the algal culture added to the experimental tank. The increase in ambient pH was 1.1 ± 0.3 (from 7.9 ± 0.2 to 9.0 ± 0.3) (median ± SD) for the fish that were exposed to *P. parvum* and 0.8 ± 0.4 (from 7.7 ± 0.3 to 8.5 ± 0.4) (median ± SD) for the fish exposed to *R. salina*. There was no significant difference in the rise in pH that was experienced by the fish exposed to either *P. parvum* or *R. salina* (t-test, *p* = 0.20). Being similar, the increase in pH is excluded as a statistical factor between the exposures.

Increased mucus shedding was observed at the gills during exposure to *P. parvum*. Similar mucus production was not observed during the hypoxia or the *R. salina* exposure experiments.

### 2.3.1. *Prymnesium parvum*

All seven fish that were exposed to *P. parvum* had a significant decrease in oxygen extraction. The effect was measured within minutes of exposure. The fish left the DeMeVOX (Device for Measuring Ventilation and Oxygen extraction) (Figure S1) between 79 and 191 min. post-algal exposure, with half of the fish leaving the device before 92 min. No fish died during the experiments.

The focus will mainly be on the last 10 min. before the fish left the device, as the fish had a continues decrease in oxygen extraction throughout the exposure to *P. parvum*. This is where the fish is under the most constraint, and also where comparison can be made to severe hypoxia. Respiratory data from pre-algae exposure and the last 10 min. of the experiment were not normally distributed (Shapiro–Wilk test) with W-values that ranged from 0.73 to 0.96 and all *p* values <0.01. Hence, a Wilcoxon signed rank test was applied to check for differences. Oxygen extraction significantly decreased by 51.7 ± 6.7 percentage points (median ± SD) (W = 14490, *p* < 0.0001) from pre-exposure to the last 10 min. No significant difference was found between the algal concentration and the decrease in oxygen extraction (Pearson Product Moment Correlation, r = −0.03, *p* > 0. 05). The ventilatory flow and ventilation frequency significantly increased by 6.3 ± 9.3 mL s$^{-1}$ and 4.0 ± 5.0 Breaths s$^{-1}$ (median ± SD) (V = 2729.5, *p* < 0.0001, 9657.5, *p* < 0.0001), respectively. There was no significant difference in oxygen consumption from pre-exposure to the end of the experiment (W = 7696, *p* = 0.44) (Table 1, Figure 1).

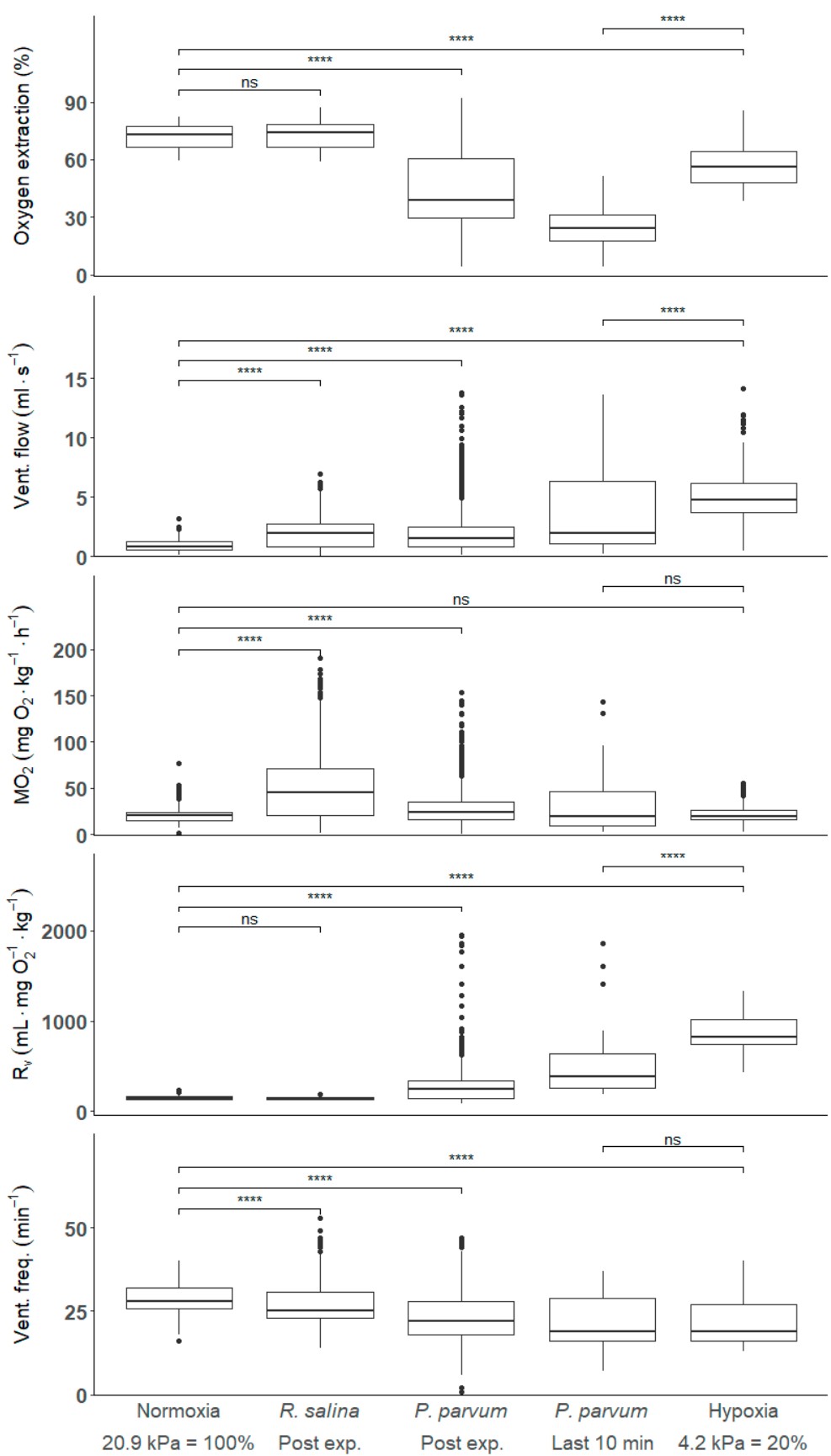

**Figure 1.** Respiratory parameters during the different stages of each experiment. All values are given as median ± SD. (**a**) Oxygen extraction, (**b**) Ventilatory flow, (**c**) Oxygen consumption, (**d**) Ventilatory requirement, and (**e**) Ventilation frequency. The significance levels are shown as asterisks (*).

### 2.3.2. *Rhodomonas salina*

The effects of *R. salina* on the fish were minimal (Table 1, Figure 1) and non-significant. The post-algal exposure measurements showed some fluctuations in the respiratory parameters, but returned to pre-exposure levels during the experiment. None of the respiratory parameters showed normal distribution from pre-algae and post-algae exposure, with W-values (Shapiro–Wilk test) ranging from 0.87 to 0.98 and all $p < 0.01$. A Wilcoxon signed rank test was used to check for differences and it showed no significant difference in any of the parameters ($E_{O_2}$: W = 134010, $p = 0.73$; $V_f$: W = 142520, $p = 0.07$; $\dot{M}O_2$: W = 137870, $p = 0.31$; $V_{freq}$: W = 140970, $p = 0.12$) (Table 1, Figure 1).

### 2.4. Ventilatory Requirement

The ventilatory requirement for normoxia and all of the exposure groups (hypoxia, *R. salina,* and *P. parvum*) were not normally distributed (Shapiro–Wilk test), with W-values ranging from 0.22 to 0.98 and all $p$ values $< 0.01$. Therefore, a Wilcoxon Rank Sum test was used to compare the exposure groups to normoxia as well as each other. Significant differences were found between all groups ($p < 0.01$) using the Bonferroni $p$ value adjustment method, except between normoxia and *R. salina* ($p = 1$) (Table 1, Figure 1).

## 3. Discussion

A favored approach of performing metabolic rate experiments on non-stressed fish is to apply intermittent stop flow respirometry [28–30]. This approach allows for the fish to acclimate to the setup over long periods and thereby eliminate most of the stress-related discrepancies in the measurements [30,31]. During the present study, measurements on oxygen extraction and ventilatory flow were desired and a different approach had to be implemented since these cannot be directly measured in a regular respirometer [9]. Therefore, a modified setup and approach used earlier [32], which allowed ventilatory flow and oxygen extraction measurements of undisturbed flatfish, was used. Preliminary experiments were conducted while using the original design [32]. European flounders (*Platichthys flesus*) that were exposed to severe hypoxia have been observed to raise their heads out of the sediment [33]. This behavior was also seen with the European plaice in the present study and it caused the measuring devices (plastic funnels) to tumble over, essentially terminating the experiment. A new measuring device, the DeMeVOX, was constructed to counter this problem based on the same principles as the original setup, but as one device with two chambers, divided by a membrane (Figure S2). During measurements at severe hypoxia, the flexibility of the membrane allowed the fish to raise their heads while keeping the two chambers separate.

The initial response in fish exposed to hypoxia is to increase ventilatory flow and to thereby increase available $O_2$ at the site of gas exchange [34]. The increase in ventilatory flow and only a slight decrease in oxygen extraction allow the fish to maintain oxygen consumption during hypoxia (termed regulator) until the oxygen levels decrease to a critical level. Thereby, the fish postpone the onset of transition from aerobic to anaerobic metabolism, essentially retaining the respiratory homeostasis [32,35,36]. Ventilatory flow needs at least to be doubled every time the $pO_2$ in the ambient water is halved to maintain the respiratory homeostasis [32]. This response was also apparent in the present study, where the increase in ventilatory flow from normoxia to severe hypoxia (20% air saturation) increased, on average, approximately six times (Table 1). A decrease in oxygen extraction usually accompanies an increase in ventilatory flow. Possible causes for the observed decrease are likely due to several interrelated reasons, such as; reduced $pO_2$ gradient at the gas exchange site between the venous blood and the inspired water [37], reduced contact time between the inspired water and the secondary lamellas [38,39], among others. This effect was also apparent in the present study, were the oxygen extraction decreased by 15.6 percentage points from normoxia to severe hypoxia (Table 1).

The same experiments were conducted using the non-harmful control alga (*R. salina*) to ensure that substances from *P. parvum* caused the effects on the fish and not by nontoxic variables. The nontoxic

variables that could affect the fish were the increased pH level or the physical appearance of algal cells in the setup. The increased pH levels after the introduction of algae were not significantly different, but the cell concentrations that the fish were exposed to were significantly different. It should be noted that a higher pH result in increased toxicity was caused by *P. parvum* [40,41]. A response elicited by the cell concentration alone should be more pronounced during the *R. salina* experiments as the fish exposed to *R. salina* experienced a higher concentration of algal cells than fish exposed to *P. parvum* (mean ± SD; $5.6 \times 10^4 \pm 6.2 \times 10^4$ cells mL$^{-1}$). The response during exposure to *P. parvum* is unlikely to be caused by a physical encounter with the algal cells, since no significant differences were found in the physiological response between pre-and post-algal exposure to *R. salina*.

In the present study, it is suggested that the effect of exposure to the ichthyotoxic alga, *P. parvum*, resemble the effects that are observed during severe hypoxia. From pre-exposure to the last 10 min, on average the ventilatory flow increased by a factor of 3.7, despite the ambient $pO_2$ being above 20 kPa (>90% air saturation), comprising a substantial increase, although not nearly the increase seen in fish exposed to severe hypoxia. The ventilation frequency for both of the treatments also showed a similar trend, in that the average frequency significantly decreased during exposure, which suggests that the decreasing efficiency of the gills caused by exposure to *P. parvum* and severe hypoxia leads to regulation in ventilatory flow through breath volume rather than frequency.

There was no significant difference in the oxygen consumption during the last 10 min. of exposure, since oxygen extraction decreased severely, even though the results from the *P. parvum* experiments showed a significant increase in ventilatory flow. However, the major difference being that for *P. parvum* exposed fish, the regulatory effort was significantly increasing the ventilation trying to maintain the oxygen uptake, even in normoxic water, again indicating the direct effect of *P. parvum* on the ventilatory requirement of fish.

An increased ventilatory flow usually is accompanied by a decrease in oxygen extraction, as shown in the hypoxia experiments. This was also the case when the fish were exposed to *P. parvum*; however, to a much larger extent, and not being caused by a decrease of the oxygen gradient across the gills. The average decrease in oxygen extraction between pre-exposure and the last 10 min. was approximately three times greater than in the average decrease from normoxia to severe hypoxia. This is a considerable difference between the two types of exposure. The magnitude of the decrease in oxygen extraction suggests the efficiency of the gills is deteriorating through *P. parvum* exposure, followed by an increase in ventilatory flow. A likely reason for the decreasing efficiency would be the formerly suggested damage to the gill epithelia (decrease effective area) and the build-up of mucus on the gills and secondary lamellae (increased diffusion distance) [10]. Inducing gill epithelial damage in rainbow trout (*Oncorhynchus mykiss*) while using zinc sulfate has shown to decrease oxygen extraction [21]. An increased mucus production can have the same effect on oxygen extraction, as demonstrated in a study where mucus acts as a diffusion barrier to oxygen [22]. An increase in the mucus layer thickness will increase the diffusion distance and reduce the diffusive transport rate of oxygen. Similar results have been obtained by Ishimatsu et al. [42] and Hishida et al. [43] in studies where Yellowtail (*Seriola quinqueradiata*) was exposed to the harmful alga *Chattonella marina*. They found that an increased mucus production at the gills causes a decrease in $pO_2$ in the arterial blood during algal exposure.

A more direct comparison between normoxia and each of the exposure treatments using the ventilatory requirement (Figure 1, Table 1) showed that there was no significant physiological effect between normoxia and exposure to *R. salina*. This finding is indictive of no effects of the nontoxic variables and that the significant difference between normoxia and exposure to *P. parvum* is likely caused by substances that are present in the culture other than the cells. Furthermore, the significant difference between the last 10 min. of exposure to *P. parvum* and severe hypoxia indicates that the effects of the toxic alga are not the same as for when the fish are exposed to severe hypoxia, neither mechanistically nor statistically.

During the present study, we did observe increased mucus production during *P. parvum* exposure by observations. A medical micro video camera was inserted into the operculum cavity after the

completion of the experiment to verify our observations. The footage confirmed that there was a greater visual amount of mucus on the gills in fish that were exposed to *P. parvum,* as opposed to fish that were exposed to *R. salina* and hypoxia (Video S1). We observed mucus on the gills of all fish exposed to *P. parvum* and the data from these experiments indicate that the fish die from suffocation when they are exposed to *P. parvum*. However, since the reduction in oxygen extraction can be seen minutes post exposure, the cause of suffocation is likely a combined effect of gill epithelial destruction and mucus acting as a barrier for oxygen transport. We have previously [9] observed fish mortality at relatively high algal concentrations without any mucus being located at the gills; this could indicate that, at relatively high *P. parvum* concentrations, fish suffocate due to gill destruction and, the fish has prolonged lifetime and suffocate due to mucus production at lower concentrations. Thus, the mode of action is likely a balance of the level of exposure.

## 4. Materials and Methods

All of the animal procedures were in agreement with the EU Directive 2010/63/EU for animal experiments and performed with permission from the Danish Animal Experiments Inspectorate (License number: 2012-15-2934-00657).

### 4.1. Experimental Fish

The fish used as a model in this study to explore the ichthyotoxic effects of *P. parvum* was European plaice (*Pleuronectes platessa*) with an average mass of 258 ± 44 g (mean ± SD). Since the blooms of *P. parvum* affect all of the fish species [5,6], the selection of the model species was based on experimental criteria, as opposed to likelyhood of the species being affected in nature. The natural behavior of the plaice is to staying buried during daytime and being active at night. This behavior allows for the non-invasive measurements of gill ventilation, oxygen extraction, and hence oxygen consumption in non-stressed fish.

The fish were caught by trawl in the Oresund strait and were transported to the Marine Biological Laboratory, University of Copenhagen, Elsinore, Denmark, and kept in fully aerated ($O_2 > 95\%$), recirculated, and filtered seawater in a 1000 L circular holding tank (10 °C, salinity approx. 30). The bottom of the holding tank was covered with 5–10 cm of sand, which allowed the fish to bury and hence reduce stress. The light:dark regime was 12:12 h.

The fish were acclimated to these conditions for a minimum of two weeks before the experiments. The fish were fed daily with trout pellets during the acclimation period (Ecolife Pearl 867, 4.5 mm, BioMar). After the acclimation period, fish were separated in small groups to another 1000 L tank without sediment allowing for visual observations of the fish health status. The fish were held in the secondary tank for a minimum of one week before the experiments, during which no feed was administered to avoid any postprandial effect in the experiments [44].

### 4.2. Algal Culture Conditions

*Prymnesium parvum* (UTEX-2797) was grown in 10 L Pyrex®bottles at 10 °C in natural seawater, with a salinity of 30, to mimic the environmental conditions in which the fish were acclimated. Before culturing, the water was heated to 95 °C for 90 minutes and subsequently cooled, before the nutrients were added to make F/2 medium [45]. The light:dark regime was 12:12h and light was provided by cool white fluorescent tubes at 120 $\mu$mol photons m$^{-2}$ s$^{-1}$. The light level was measured inside the algal culture bottle with a Li-Cor®, LI-1000 radiation sensor that was equipped with a spherical probe. Aeration (~6 L h$^{-1}$) was provided through glass tubes with a diameter of 4 mm. The algal cultures were inoculated to a starting concentration of $10^4$ cells mL$^{-1}$ and grown to a concentration of $463 \times 10^3 \pm 134 \times 10^3$ cells mL$^{-1}$ (median ± SD) before use. All of the algal cultures that were used for experiments were in the exponential growth phase. The non-toxic algal specie that was utilized in the control experiments, *R. salina*, was grown under the same conditions.

### 4.3. Device for Measuring Ventilation and Oxygen eXtraction (DeMeVOX)

The DeMeVOX is an improved version of the experimental setup that was described by Kerstens et al. [32], by which the oxygen extraction and ventilation volume of flatfish were measured. The device consists of two chambers that were constructed from clear acrylic glass and separated by a soft membrane made of a vinyl glove (Ecoline Plus, Ecoline GmbH) (Figure S2).

The front chamber of the DeMeVOX was fitted with a Gould-Statham electromagnetic blood flow probe (diameter 8 mm, length 50 mm), which was connected to a Gould-Statham blood flow meter (type SP 2202, Statham Instruments, Oxnard, CA, USA). The flow meter operated with a sampling frequency of 200 Hz. The flow meter continuously measured the gill ventilatory flow and breathing frequency during the experiments. The analog output from the flow meter passed to a real-time data-acquisition system (MP100, Biopac Systems, Goleta, CA, USA) and monitored in AcqKnowledge (version 3.9.1.6, Biopac Systems, Goleta, CA, USA). The flowmeter was calibrated while using a beaker, a stopwatch, and one meter of silicone tubing with continuous water flow using a siphoning system. Changing the height of the outflow and adjusting the flow rate made linear calibration possible ($r^2 = 0.99$). The equation of the regression line was used in the AcqKnowledge software to convert the mV output to mL min$^{-1}$. pO2 was measured in the rear chamber of the DeMeVOX and the ambient water to measure the oxygen tension in the inspired and expired water, respectively, by using fiber optic sensor technology (Fibox 3, PreSens, Regensburg, Germany) that was connected to the Biopac system. The tightness of the DeMeVOX seal was tested in several experiments during preliminary measurements with and without fish. The pilot test was conducted by injecting red food color into the front chamber and along the edges of the DeMeVOX.

### 4.4. Experimental Setup

The experiments were carried out in an acrylic tank with a 5 cm layer of sand covering the bottom ($55 \times 46 \times 19.5$ cm; water volume without and with sand was 49.3 and 36.7 L, respectively). The water in the system was filtered using a UV filter and the temperature was maintained at $10 \pm 1.0$ °C using a temperature regulator (PR5714, PR-Electronics, Ronde, Denmark), a cooler (Hetofrig, Birkerod, Denmark), and a stainless-steel cooling coil. Before and between the experiments, a 100 L recirculation pump sump with seawater was connected to the setup, to ensure high buffer capacity to counter any accumulation of ammonia and $CO_2$.

The fish were allowed to acclimate in the experimental tank for a minimum of 24 h before the experiments [46,47]. The hypoxia and algal exposure experiments were conducted on one fish at the time. When the fish was buried in the sediment, only the mouth, eyes, and edge of the upper operculum were visible. The DeMeVOX was placed on top of the fish before the experiments, so the membrane was situated behind the eyes and in front of the operculum (Figure S1). The DeMeVOX was gently pushed down to complete the seal, so the membrane and the sediment on top of the fish separated the two chambers. Once measurements of oxygen extraction and ventilation were stable, the experimentation started. Each fish was initially exposed to hypoxia and after a recovery period of minimum 48 h, seven fish were exposed to *P. parvum,* and the remaining seven fish were exposed to the non-toxic control alga *R. salina*. After each algal exposure, the fish was removed and euthanzied, and the setup was emptied and cleaned thoroughly (Figure S3). The fish that were exposed to *P. parvum* and *R. salina* had an average mass of $252 \pm 56$ and $264 \pm 30$ g (mean ± SD), respectively.

After experimental completion, a medical micro video camera (VivasightTM, ET view Ltd, Misgav, Israel) was inserted into the operculum cavity of the fish to visually observe the gills for any effects of the exposures (Video S1).

The fish could leave the DeMeVOX as desired during the severe hypoxia and algal experiments. To avoid any unnecessary suffering of the fish, escape from the DeMeVOX was considered to be the humane endpoint and thereby the termination of the algae exposure treatments.

4.4.1. Severe Hypoxia Experiments

The pO2 level in the ambient water was measured by a fiber-optic oxygen sensor (Fibox 3, PreSens, Regensburg, Germany) and regulated by bubbling nitrogen into the experimental tank through an air stone.

The severe hypoxia experiments were conducted by lowering the pO2 level in the ambient tank in six steps (20.9, 16.8, 12.6, 8.4, 6.3, 4.2 kPa) for one hour at each level to simulate natural conditions. Furthermore, the stepwise hypoxia method was used to increase the likelihood of the fish not leaving the DeMeVOX before reaching the target pO2 level of 4.2 kPa. The oxygen concentration at 4.2 kPa is 1.9 mg $O_2$ $L^{-1}$ or 20% air saturation and considered severe hypoxia. Eight fish were chosen for further analysis, as they completed one-hour of measurements at 4.2 kPa i.e. severe hypoxia, during the hypoxia exposure.

4.4.2. *P. parvum* Experiments

To the author's knowledge, no direct measurements have been published regarding the binding properties of the suspected toxins, the prymnesins, to polymeric surfaces (experimental tank), although a computer model did show this to be possible [40]. If this is the case, this will affect the toxicity, but the exposure concentrations that we employed were high enough to elicit a physiological response and within ecological relevant concentrations measured at natural blooms [1].

The algal exposure experiments were conducted at a minimum of 48 h post hypoxia experiments by which time any excess post-hypoxic oxygen consumption (EPHOC) would have ceased [48–50]. Disconnecting both the UV filter and the 100 L recirculation barrel from the setup during algal exposure experiments mitigated algal death and dilution of the toxins.

The DeMeVOX was placed on the top of the fish and control measurements taken for two hours before the introduction of algae, similarly to during the hypoxia experiments. The pH of the tank water was continuously measured (WTW pH 3210, Weilheim, Germany). Each experiment used 20 L of algal culture, and therefore the equivalent volume of water had to be removed from the setup before the algae could be introduced. The algal cultures were slowly poured into the setup ensuring a mixture of algae and the water already present in the setup. Before the introduction of algae to the setup, algal concentrations of the culture were established while using a Coulter Counter Multisizer 3 (Beckman Coulter, High Wycombe, UK). To allow for mixing in the experimental setup, pH and algal concentration were measured 15 min. after the addition of the algae. The average concentration of *P. parvum* in the setup was $2.64 \times 10^5 \pm 0.6 \times 10^5$ cells $mL^{-1}$ (mean ± SD).

4.4.3. *R. salina* Experiments

Experiments using the control alga, *R. salina*, were conducted using the same approach as the *P. parvum* experiments. The average mixed concentration of *R. salina* in the setup was $3.2 \times 10^5 \pm 0.7 \times 10^5$ cells $mL^{-1}$ (mean ± SD). All of the *R. salina* exposure experiments were terminated 180 min. post algal exposure.

*4.5. Data Analysis and Statistics*

All data was analyzed using the statistical software R (Version 3.2.3) and tested for normality and homoscedasticity using a Shapiro–Wilks test and an F-test, respectively. The statistical tests for differences were chosen based on the results of these tests and the data structure.

Since a complete data set from the severe hypoxia exposure could not be obtained from all fourteen fish, eight fish were chosen for further analysis, representing fish from both groups that were exposed to *P. parvum* and *R. salina*. Differences in body masses between the three groups were statistically tested (*R. salina*, *P. parvum*, and hypoxia groups).

The DeMeVOX data was divided into periods for each of the exposure types. Thirty min. periods of stable ventilation in normoxia and hypoxia were analyzed in the hypoxia experiments. In the

*R. salina* experiments, 30 min. and 180 min. periods were analyzed from the pre-algal and post-algal introduction, respectively. As the fish eventually escaped in the *P. parvum* experiments, 30 min. periods from pre-algal were analyzed, and for the post-algal introduction, all of the measurements were analyzed until the fish left the DeMeVOX (79–191 min). For a comparison between pre- and post-algal introduction, only the last 10 min. before the fish left the DeMeVOX are used and will be referred to as the last 10 min.

The analysis was based on calculations of each breath and averaged per min. The total ventilatory volume (mL) was determined by integrating the ventilatory flow rate (mL s$^{-1}$). The oxygen extraction from the ventilatory flow across the gills, as given in percent, was calculated using the following equation:

$$E_{O_2}(\%) = (pO_{2\,I} - pO_{2\,E}) \cdot pO_{2\,I}^{-1}, \tag{1}$$

where $E_{O_2}(\%)$ is the oxygen extraction, $pO_{2I}$ the oxygen tension of the inspired ambient water, and $pO_{2E}$ the oxygen tension of the expired water, as measured in the rear chamber of the DeMeVOX.

Oxygen consumption (MO$_2$) was calculated according to the Fick principle, by applying the ventilation volume data and the average oxygen extraction using the following equation:

$$MO_2 = pO_2 \cdot \beta_{O2} \cdot E_{O_2} \cdot V_f \cdot m_f^{-1} \cdot 10^{-2}, \tag{2}$$

where MO$_2$ is the oxygen consumption (mg O$_2$ kg$^{-1}$ h$^{-1}$), pO$_2$ equals the partial pressure of oxygen in the water (kPa), $\beta O2$ equals the oxygen solubility of the water at a certain temperature and salinity (mg O2 L$^{-1}$/mm pO$_2$), Eo$_2$ equals the oxygen extraction (%), V$_f$ equals the ventilatory flow (mL h$^{-1}$), and mf equals the body mass of the fish (kg). Ventilation frequency was determined while using the pulsatile ventilation wave and was calculated for each min. Ventilatory requirement was used in order to compare the different treatment groups against each other. Ventilatory requirement is the ratio between the ventilatory flow and the oxygen consumption and describes how many L of water a fish must ventilate per kg per acquired mg O$_2$. The ventilatory requirement was calculated while using the following equation:

$$R_V = V_f \cdot MO_2^{-1}, \tag{3}$$

where R$_V$ is the ventilatory requirement (L mg O$_2$$^{-1}$ kg$^{-1}$), V$_f$ is the ventilatory flow (L h$^{-1}$), and MO2 is the oxygen consumption (mg O$_2$ kg$^{-1}$ h$^{-1}$).

All data that is not normally distributed and has outliers will be shown as a median ± SD, and the rest will be shown as mean ± SD.

**Supplementary Materials:** The following are available online at http://www.mdpi.com/2410-3888/4/2/32/s1, Figure S1: Illustration of DeMeVOX in use, Figure S2: DeMeVOX blueprints, Figure S3: Experimental flow chart, Video S1: Post experiment footage of the gills in four different fish, 3D File S1: 3D printer friendly version of the DeMeVOX's front chamber, 3D File S2: 3D printer friendly version of the DeMeVOX's read chamber.

**Author Contributions:** H.B., N.R.A., P.J.H. and J.F.S. conceived the study. H.B. and N.R.A. carried out the experimental work. H.B., N.R.A. and M.B.S.S. performed the analysis and interpretations. H.B., N.R.A., M.B.S.S., P.J.H. and J.F.S. contributed to the writing and finalizing the paper.

**Funding:** This research was funded by the Danish Council for Strategic Research through the project "HABFISH", Project No. 0603-00449B.

**Conflicts of Interest:** The authors declare no conflict of interest.

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
