# Peer review of "Respiratory Physiology of European Plaice (Pleuronectes platessa) Exposed to Prymnesium parvum"

_fishes, doi:10.3390/fishes4020032_

Round 1
Reviewer 1 Report
p.p1 {margin: 0.0px 0.0px 0.0px 0.0px; font: 12.0px Helvetica} p.p2 {margin: 0.0px 0.0px 0.0px 0.0px; font: 12.0px Helvetica; min-height: 14.0px} p.p3 {margin: 0.0px 0.0px 0.0px 0.0px; font: 13.3px Times; color: #000000} p.p4 {margin: 0.0px 0.0px 0.0px 0.0px; font: 12.0px Times; color: #000000} p.p5 {margin: 0.0px 0.0px 0.0px 0.0px; font: 12.0px Times; color: #000000; min-height: 14.0px} span.s1 {font: 12.0px Helvetica; color: #000000} span.s2 {font-kerning: none} span.s3 {font: 16.0px 'Times New Roman'; font-kerning: none} span.s4 {font: 13.3px Times; font-kerning: none}Introduction:
L 39. Not a full sentence (“Hypothesized[…]”
L 40: Toxin(s) . One or multiple?
L 45-40: Aims slightly unclear. Normal conditions vs. severe hypoxia? Please make clearer. (Becomes clear when reading Results or M&M, but please rephrase to make the objective more easily graspable (this could also be improved in your abstract).
Material and Methods:
L281: How long were they exposed to hypoxia?
L281: What is the n for each stage? 7 fish hypoxia, 7 fish hypoxia —> P. Parvum? And 7 fish hypoxia, R. Salina? The paper would benefit from an experimental flow chart (even if only in the supplementary materials).
L282: How long were they exposed to the algae?
290: How many remained till the end of the experiment and how many, how many left prematurely?
Results:
L59: Despite reading the M&Ms first I am still confused by the experimental setup. Either add a flowchart, or make the setup more clear (also N= would be interesting for each experiment)
L65: decreased significantly compared to what?
L71: Why are there different measurements for R. Salina and P. Parvum (post exp. and last 10min?) Is the Normoxia section in the Hypoxia exposure treatment pre or post? You could indicate in the table which ones are significantly different.
L77: Grammatically incorrect. Rephrase.
L77: … wich showed a significant difference between the two exposure concentrations. Significant difference in what? Concentration of the 2 different algae? In my opinion the control (R. Salina) should have the same concentration as the treatment, to act as a proper control. It says different exposure concentrations, but I am not sure which exposure concentrations you are talking about?
L94: also < 0.001? Is there a table somewhere where all p values are given?
L116: Which stages for R. Salina and P. parvum? I assume post? And for Hypoxia?
Discussion:
L132: Where is Figure 2? It’s not in the MS.
L153: Ok, this answers my question from before. But it would be nice to have this information already earlier in the text. I’d suggest to point out, that the control has a higher concentration, than the treatment.
L171: Did all fish survive the treatment? Or did some perish after?
L176: Is this plowable, not as a bar plot but a scatter plot (e.g. the ratio of ventilatory flow against oxygen extraction over time, for both treatments (parvum, hypoxia) as well as control.). In my opinion this would add to your findings and also would take out doubts over the slight ambiguity of the choosing of the endpoints (see table 1, pre exposure, post vs. last 10 min).
L208: Rephrase this sentence ( I think you are missing a subject).
211: Again, did any (all) of these fish die/suffocate post exposure? Would be an interesting point.
Author Response
Thank you for your comments. You can find our answers in the attached file.

Reviewer 2 Report
Line 59. This belongs to the methods. Table 1. Indicate with an asterisk the groups that are significantly different. Line 95. This is not the value in the table. Also, reference the results to Table 1. When detailing results indicate if they are related with table 1 or figure 1. Also, show the average and standard deviation values for the figure 1 in the text. I cannot find figure 2. The discussion starts with a description of the respirometry methods; there was not mention of respirometry methods in the introduction. This paragraph feels disconnected from the full story. The discussion does not follow a logic flow, the discussion need to be re-written following the hypothesis and highlighting the main conclusions. There is no reference to the supplementary figures in the main text.Author Response
Thank you for your comments. You can find our answers in the attached file.

Reviewer 3 Report
This study examined the impacts of P. parvum on respiratory physiology of European plaice. The results showed that the presence of P. parvum, presumably via toxins produced by this alga, causes irritation of the gill epithelium with consequent increase in mucus production, and thus impairs the fish's ability to extract oxygen from water. Exposure to a non-toxic alga had no effects on the gills. As previously concluded for rainbow trout by the same group of investigators, the conclusion of this study is that exposure to P. parvum causes suffocation in plaice.
The experimental design and analytical procedures are straightforward and conclusions are reasonable. My only substantive comment, which I would regard as a moderate concern, is about the choice of model species and experimental salinity. In nature, P. parvum rarely, if ever, blooms and causes fish kills in seawater. Most toxic blooms occur at the low end of the brackish-salinity range. Also, to my knowledge, plaice is not a species that has been affected by P. parvum blooms in nature. The authors should provide rationale for (1) the study species - being a model species is a good thing, but what else specifically can be gained by using plaice? And (2) the use of seawater salinity levels (30).
Author Response

(The authors gave the same response as above.)
